# Neighbor-aware Contrastive Disambiguation for Cross-Modal Hashing with Redundant Annotations

**Chao Su [1], Likang Peng [1], Yuan Sun [2], Dezhong Peng [1,3], Xi Peng [1,2], Xu Wang [1,4]\***

[1]College of Computer Science, Sichuan University, Chengdu, China
[2]National Key Laboratory of Fundamental Algorithms and Models
for Engineering Numerical Simulation, Sichuan University, Chengdu, China
[3]Tianfu Jincheng Laboratory, Chengdu, China
[4]Centre for Frontier AI Research (CFAR), A*STAR, Singapore
suchao.ml@gmail.com, likangpeng@stu.scu.edu.cn, sunyuan_work@163.com,
pengdz@scu.edu.cn, pengx.gm@gmail.com, wangxu.scu@gmail.com

## Abstract

Cross-modal hashing aims to efficiently retrieve information across different modalities by mapping data into compact hash codes. However, most existing methods assume access to fully accurate supervision, which rarely holds in real-world scenarios. In fact, annotations are often redundant, i.e., each sample is associated with a set of candidate labels that includes both ground-truth labels and redundant noisy labels. Treating all annotated labels as equally valid introduces two critical issues: (1) the sparse presence of true labels within the label set is not explicitly addressed, leading to overfitting on redundant noisy annotations; (2) redundant noisy labels induce spurious similarities that distort semantic alignment across modalities and degrade the quality of the hash space. To address these challenges, we propose that effective cross-modal hashing requires explicitly identifying and leveraging the true label subset within all candidate annotations. Based on this insight, we present Neighbor-aware Contrastive Disambiguation (NACD), a novel framework designed for robust learning under redundant supervision. NACD consists of two key components. The first, Neighbor-aware Confidence Reconstruction (NACR), refines label confidence by aggregating information from cross-modal neighbors to distinguish true labels from redundant noisy ones. The second, Class-aware Robust Contrastive Hashing (CRCH), constructs reliable positive and negative pairs based on label confidence scores, thereby significantly enhancing robustness against noisy supervision. Moreover, to effectively reduce the quantization error, we incorporate a quantization loss that enforces binary constraints on the learned hash representations. Extensive experiments conducted on three large-scale multimodal benchmarks demonstrate that our method consistently outperforms state-of-the-art approaches, thereby establishing a new standard for cross-modal hashing with redundant annotations. Code is available at https://github.com/Rose-bud/NACD.

## 1 Introduction

With the explosion of large-scale and diverse data on the Internet [1–7], efficiently retrieving semantically relevant data across modalities has become increasingly important [8–16]. For large-scale datasets, cross-modal hashing (CMH) offers an effective solution by encoding heterogeneous data into compact binary hash codes, enabling high retrieval efficiency and low storage cost. The core

---

*Corresponding author.

39th Conference on Neural Information Processing Systems (NeurIPS 2025).

challenge of CMH lies in effectively leveraging available supervision while minimizing semantic discrepancies between different modalities.

Existing CMH methods can be broadly categorized into unsupervised and supervised approaches based on whether the label information is available. The unsupervised CMH methods [17–22] learn hash functions by exploring the intrinsic structure and similarity of the data without access to labels. For instance, CIRH [21] jointly preserves the multimodal correlation and identity semantics into binary hash codes based on a heterogeneous graph network. Moreover, UCCH [22] proposes a contrastive learning-based unsupervised CMH method with a momentum optimizer and cross-modal ranking learning loss to improve performance. However, the lack of supervision limits their ability to learn semantically discriminative representations. In contrast, supervised methods [23–31] leverage label information to learn more discriminative hash codes to improve retrieval performance. Within a probabilistic modality alignment framework, MIAN [27] investigates the preservation of asymmetric similarities both within and between modalities, thereby fully utilizing multi-level semantic information throughout the entire database. RSHNL [31] designs a Robust Self-paced Hashing mechanism that mitigates the misleading effects of noisy labels on the model by simulating the human cognitive process. While these supervised methods have achieved satisfactory retrieval performance by leveraging label information, they implicitly rely on two assumptions: (1) all labels in the training data are accurate; (2) noisy annotations are simulated by replacing correct labels with incorrect ones. However, in real-world applications, data annotations are often redundant, i.e., each instance is labeled with a candidate label set that includes both true and spurious labels. We refer to this setting as redundant annotations. As shown in Fig. 1, among the annotations of the anchor sample pair, only "Sea, Plant, Beach, Cloud" are correct labels, while the others are additional noisy labels. Such redundancy can severely distort semantic similarity estimation and hinder effective hash learning by introducing spurious correlations across modalities. Crucially, existing methods fail to explicitly distinguish true labels within the candidate label set, leading to degraded performance under redundant noisy supervision. Although partial multi-label learning (PML) [32–35] provides a potential solution for redundant annotations, most PML methods assume a shared feature space and overlook cross-modal semantic divergence. In contrast, CMH must address both accurate label disambiguation and modality-robust contrastive pair construction, which remains a rarely explored challenge.

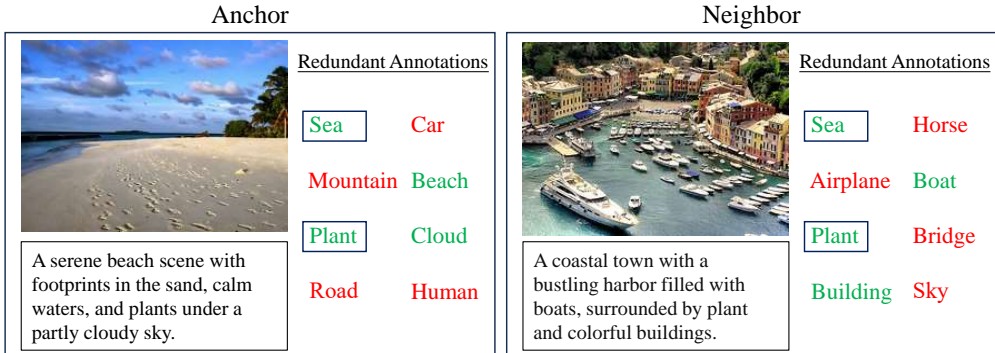

Figure 1: The redundant annotations in cross-modal hashing. Among the annotations, the correct labels and redundant noisy labels are represented by green and red, respectively. Shared labels between the anchor sample pair and its neighbor sample pair are enclosed in dashed boxes.

In this paper, we focus on the practical scenario of redundant annotations and propose a novel framework, Neighbor-aware Contrastive Disambiguation (NACD), for robust cross-modal hashing. NACD benefits from an efficient Neighbor-aware Confidence Reconstruction (NACR) module and a novel Class-aware Robust Contrastive Hashing (CRCH) module. Specifically, NACR integrates sample label confidence with its cross-modal neighborhood label confidence to identify the ground-truth labels within the entire annotations. CRCH dynamically constructs positive and negative sample pairs based on class confidence thresholds, reducing erroneous associations caused by misleading labels and significantly enhancing the model's robustness to redundant supervision. Moreover, to effectively reduce the quantization error, we employ an effective quantization loss that enforces binary constraints on the learned hash representations. The main contributions of the proposed NACD are as follows:

- We first focus on the practical scenario of redundant annotations in cross-modal hashing and propose a novel Neighbor-aware Contrastive Disambiguation (NACD) framework that addresses the challenges caused by redundant noisy supervision.

- To achieve accurate label disambiguation, an efficient Neighbor-aware Confidence Reconstruction (NACR) module is presented, which integrates the label confidence of the anchor sample pair with that aggregated from its cross-modal neighbors to identify the ground-truth labels within the entire annotations.

- We design an innovative Class-aware Robust Contrastive Hashing (CRCH) module which dynamically constructs positive and negative sample pairs based on class-wise confidence thresholds, reducing erroneous associations caused by redundant noisy labels and significantly enhancing the model's robustness to incorrect supervision.

- Comprehensive experiments on three multimodal datasets, i.e., MIRFlickr-25k, NUS-WIDE, and MS-COCO, demonstrate that NACD consistently outperforms state-of-the-art CMH methods across various redundancy levels.

## 2 Related Work

### 2.1 Cross-Modal Hashing

Cross-modal hashing aims to retrieve semantically relevant data across different modalities within a shared Hamming space. The key challenge lies in bridging the modality gap. To address this challenge, numerous approaches have been proposed, which can be broadly divided into two categories: unsupervised CMH methods and supervised CMH methods. More specifically, unsupervised CMH methods [19–22] learn modality-specific transformations by maximizing cross-modal correlations without label supervision. For example, UCCH [22] integrates contrastive learning into unsupervised CMH to enhance retrieval performance and robustness. However, these unsupervised methods suffer from limited performance due to the absence of explicit supervision. Supervised CMH methods [29–31, 36] are typically based on two assumptions: (1) all labels in the training data are accurate; (2) noisy annotations are simulated by replacing correct labels with incorrect ones. For example, HCCH [36] introduces a coarse-to-fine hierarchical hashing strategy to effectively utilize hierarchical features and accurate labels across modalities. RSHNL [31] proposes a robust self-paced hashing mechanism that emulates human cognition, thereby reducing the negative impact of noisy labels and improving model performance.

However, in real-world applications, multimodal annotations often contain "redundant annotations". Therefore, this paper focuses on a largely unexplored yet challenging problem: cross-modal hashing with redundant annotations.

### 2.2 Learning with Redundant Annotations

Partial multi-label learning (PML) trains models using redundantly annotated data, where each instance is associated with a candidate label set containing both true and redundant noisy labels. The key challenge in PML lies in filtering out incorrect labels and identifying reliable ones, thereby recovering the true label distribution for supervision. To achieve this, a number of methods have been developed. These methods can be broadly categorized into smoothness assumption-based, low-rank constraint-based, and sparsity regularization-based approaches. Smoothness assumption-based approaches [35, 37, 38] are based on the assumption that neighboring samples in the feature space are more likely to have similar labels. The low-rank constraint-based approaches [39–41] leverage the low-rank property to achieve disambiguation. The sparsity regularization-based approaches [32, 33, 40] impose sparsity on the candidate label set, effectively suppressing noisy labels and facilitating disambiguation.

In contrast to the aforementioned PML methods, our approach integrates the label confidence of an anchor sample pair with that aggregated from its cross-modal neighbors to identify true labels within all annotations. Furthermore, a dynamically updated class-wise threshold enables more accurate and adaptive label disambiguation.

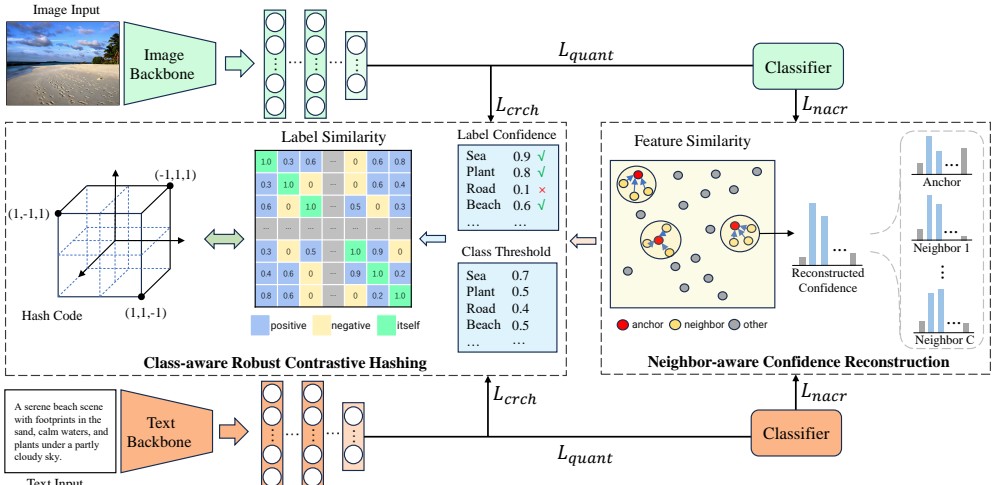

Figure 2: The pipeline of the proposed framework NACD for cross-modal hashing with redundant annotations. NACR refines label confidence by aggregating information from cross-modal neighbors to distinguish true labels from redundant noisy ones. Meanwhile, CRCH constructs reliable positive and negative pairs based on the learned label confidence, which significantly improves robustness against noisy supervision.

## 3 Proposed Approach

### 3.1 Problem Definition

For ease of presentation, we first give some definitions of cross-modal hashing of redundant annotations. Suppose the input space is denoted as $\mathcal{X}$, and the label space as $\mathcal{Y} = \{1, 2, ..., K\}$, where $K$ indicates the total number of classes. Denote $\left\{ \left\{ x_j^m \right\}_{m=1}^2, y_j \right\}_{j=1}^N$ as the training set with $N$ sample pairs, where $x_j^m$ represents the $j$-th instance from the $m$-th modality ($m = 1$ for image and $m = 2$ for text). $y_j \in \{0, 1\}^K$ denotes the candidate label vector of the $j$-th sample pair, which encodes both ground-truth and redundant noisy labels. The $k$-th element of $y_j$ equals 1 if the corresponding sample pair is annotated as class $k$. Here, $p_j$ is the label confidence vector of the candidate label vector $y_j$. The hash codes are denoted as $\left\{ \boldsymbol{b}_j^m \right\}_{j=1}^N \in \{-1, 1\}^L$, where $L$ is the hash code length. CMH leverages hash functions to map data from different modalities into a common Hamming space, enabling efficient similarity search across modalities through compact binary codes. Let the hash functions be $f^m$, where $m \in \{1, 2\}$. Due to the NP-hard problem in binary optimization, we calculate the hash representations by $h_j^m = tanh(f_m(x_j^m)), m \in \{1, 2\}$ in the training process. Thus, the final binary hash codes are obtained by applying the $sign$ function: $b_j^m = sign(h_j^m), m \in \{1, 2\}$. Additionally, a linear classifier with a sigmoid activation function $g(\cdot)$ is employed to obtain the probability distribution $z_j^m = g(h_j^m)$, where $m \in \{1, 2\}$.

### 3.2 Neighbor-aware Confidence Reconstruction

Under redundant supervision, confidence estimates obtained from individual samples may be unreliable. Meanwhile, semantically similar samples across modalities often share the same true labels. Motivated by this observation, we present a confidence-mixture (CM) strategy to reconstruct label confidence by balancing neighborhood consensus and self-prediction.

First, assume that the prediction probability of the $j$-th target sample $\left\{ x_j^m \right\}_{m=1}^2$ is $z_j$, under the supervision of redundant annotations $y_j$. We define the confidence of each sample's prediction by the model as:

$$p_j \leftarrow \gamma p_j + (1 - \gamma) \frac{1}{2} \sum_{m=1}^2 z_j^m \circ y_j, \tag{1}$$

where $\circ$ denotes Hadamard product, and $\gamma$ is a momentum parameter that decays from 0.95 to 0.8 during training.

Since model predictions may be unreliable under redundant supervision, and similar samples tend to share the same true labels, we exploit the predictions of cross-modal neighbors to refine the confidence estimation. Given the $C$ nearest neighbors $\mathcal{N}_j$ of the anchor sample pair $x_j$, the neighbor-aggregated confidence $q_j$ is calculated as:

$$q_j = \frac{\sum_{c \in N_j} s_{jc} p_c}{\sum_{c' \in N_j} s_{jc'}}, \qquad \text{where } s_{jc} = \tfrac{1}{2} \left( s_{jc}^1 + s_{jc}^2 \right), \tag{2}$$

where $s_{jc}$ denotes the fused similarity between the anchor $j$ and its neighbor $c$, obtained by averaging the similarities from the image and text modalities. To mitigate the issue that inaccurate estimates during training may hinder model optimization, we reconstruct the anchor confidence $p_j$ by mixing it with the neighbor-aggregated confidence $q_j$. The reconstructed confidence is computed as:

$$p_j \leftarrow \lambda q_j + (1 - \lambda) p_j, \tag{3}$$

where $\lambda$ is a mixture coefficient that balances neighborhood consensus and self-prediction. After obtaining reconstructed confidence by Eq. (3), the disambiguation loss $\mathcal{L}_{nacr}$ can be formulated as:

$$\mathcal{L}_{nacr} = -\frac{1}{2N} \sum_{m=1}^{2} \sum_{j=1}^{N} \sum_{k=1}^{K} \left[ p_{jk} \log \left( z_{jk}^m \right) + (1 - p_{jk}) \log \left( 1 - z_{jk}^m \right) \right], \tag{4}$$

where $z_{jk}^m$ denotes the predicted probability that the $j$-th sample belongs to the $k$-th class under modality $m$, and $p_{jk}$ represents the reconstructed confidence of the $j$-th sample pair for class $k$.

Empirically, incorporating neighborhood consensus into label confidence estimation improves disambiguation accuracy, especially with abundant redundant noisy annotations. It enables the model to construct semantically faithful contrastive pairs, thus significantly enhancing the model's robustness.

### 3.3 Class-aware Robust Contrastive Hashing

Although NACR reconstructs label confidence effectively, it does not explicitly incorporate class-level information when constructing robust positive and negative pairs. Therefore, we propose a Class-aware Robust Contrastive Hashing (CRCH) module to adaptively build reliable pseudo-labels thereby enhancing the stability of contrastive optimization.

First, we establish a class-wise threshold for each class based on the reconstructed label confidence. The threshold $t_k$ for the $k$-th class is calculated as follows:

$$t_k = \frac{1}{N_k} \sum_{j=1}^{N} p_{jk}, \tag{5}$$

where $p_{jk}$ denotes the reconstructed label confidence of the $j$-th sample pair on class $k$, and $N_k$ indicates the number of samples for which $p_{jk} > 0$ among all $N$ samples. This class-specific averaging adaptively captures the distributional characteristics of each class rather than relying on a fixed threshold. To build class-wise supervision, we derive the pseudo-label $\hat{y}_j = [\hat{y}_{j1}, \ldots, \hat{y}_{jK}]$ by comparing each reconstructed confidence $p_{jk}$ with its corresponding class-wise threshold $t_k$:

$$\hat{y}_{jk} = \begin{cases} 1, & \text{if } p_{jk} \geq t_k \\ 0, & \text{otherwise} \end{cases}, \tag{6}$$

where $\hat{y}_{jk} \in \{0, 1\}$ indicates whether class $k$ is considered positive for the $j$-th sample pair. For a mini-batch containing $n$ sample pairs, we compute a label similarity matrix $T \in [0, 1]^{n \times n}$ based on the intersection-over-union (IoU) between pseudo-label vectors:

$$T_{ij} = \frac{\hat{y}_i \cap \hat{y}_j}{\hat{y}_i \cup \hat{y}_j}, \tag{7}$$

where $T_{ij}$ measures the semantic similarity between the $i$-th and $j$-th sample pair based on their pseudo-labels. A higher $T_{ij}$ indicates stronger semantic consistency, while $T_{ij} = 0$ means that

the sample pairs are semantically disjoint. Accordingly, positive and negative pairs are determined by the indicator functions $\mathbb{I}[T_{ij} > 0]$ and $\mathbb{I}[T_{ij} = 0]$, respectively. However, directly relying on $T_{ij}$ for pair construction can be unreliable due to redundant noisy annotations. To mitigate this issue, we follow [22] and introduce a margin-based thresholding mechanism that adaptively adjusts cross-modal similarities matrix $S$ to reduce the impact of negative pairs with overly high similarities. Specifically, the adjusted similarity matrix $N_{ij}^*$ is defined as:

$$N_{ij}^* = \begin{cases} S_{ij}^*, & \text{if } S_{ij}^* \geq S_{ii} - \delta \\ S_{ij}^* - \xi, & \text{otherwise} \end{cases}, \tag{8}$$

where $* \in \{12, 21\}$ denotes image-to-text and text-to-image retrieval directions. $S_{ij}^{12} = h_i^1 \cdot h_j^2$, $S_{ij}^{21} = h_i^2 \cdot h_j^1$. The margin parameter $\delta$ distinguishes hard and easy negative pairs, while the shift parameter $\xi$ suppresses the influence of overly easy negatives. The loss of CRCH is defined as:

$$\mathcal{L}_{crch} = \frac{1}{n^2} \sum_{i=1}^{n} \sum_{j \neq i}^{n} \left[ \mathbb{I}[T_{ij} = 0] \cdot \sum^{*} \exp\left(N_{ij}^*\right) + \mathbb{I}[T_{ij} > 0] \cdot \exp(-S_{ij} - T_{ij}) \right] - \frac{1}{n} \sum_{i=1}^{n} S_{ii}, \tag{9}$$

where $\mathcal{L}_{crch}$ consists of three components: (1) $\mathbb{I}[T_{ij} = 0] \cdot \sum^{*} \exp(N_{ij}^*)$ penalizes all negative cross-modal sample pairs, especially those that are easily confusable. (2) $\mathbb{I}[T_{ij} > 0] \cdot \exp(-S_{ij} - T_{ij})$ treats cross-modal sample pairs with non-zero label similarity $T_{ij}$ as positive pairs and encourages greater semantic alignment. The higher the label similarity, the stronger the penalty imposed for insufficient feature similarity $S_{ij}$. (3) $-\frac{1}{n} \sum_{i=1}^{n} S_{ii}$ encourages consistency within each sample pair by increasing the similarity between the image and text representations.

The CRCH module dynamically constructs positive and negative sample pairs based on class-wise confidence thresholds, enabling the model to effectively capture nuanced relationships between samples. This design significantly reduces erroneous associations caused by redundant noisy labels and enhances the model's robustness to incorrect supervision.

## 3.4 Optimization

Benefiting from the NACR and CRCH modules, our model effectively learns discriminative hash codes within the Hamming space. However, the discrepancy between continuous representations and discrete binary codes inevitably leads to quantization errors, which can substantially degrade retrieval performance in CMH. To address this issue, we present an effective quantization loss as follows:

$$\mathcal{L}_{quant} = \frac{1}{n \cdot L} \sum_{m=1}^{2} \sum_{j=1}^{n} \sum_{l=1}^{L} \left| h_{jl}^m - sign(h_{jl}^m) \right|, \tag{10}$$

where $h_{jl}^m$ denotes the $l$-th element of the hash representation $h_j^m$. This loss penalizes the deviation between continuous hash values and their corresponding binary codes $\text{sign}(h_{jl}^m)$, thereby encouraging each element to approach $\pm 1$ and effectively reducing the quantization error, as demonstrated in our ablation studies. Thus, the final loss function of NACD can be defined as:

$$\mathcal{L} = \mathcal{L}_{nacr} + \alpha \mathcal{L}_{crch} + \beta \mathcal{L}_{quant}, \tag{11}$$

where $\alpha$ and $\beta$ are hyperparameters that balance the contributions of $\mathcal{L}_{nacr}$, $\mathcal{L}_{crch}$, and $\mathcal{L}_{quant}$. Additional optimization details of NACD are provided in Appendix Sec. A.

## 4 Experiments

### 4.1 Experimental Settings

To evaluate the effectiveness of NACD, we conducted experiments on three benchmark datasets: MIRFlickr-25k (Flickr) [42], NUS-WIDE (NUS) [43], and MS-COCO (COCO) [44]. For all methods, we evaluate the Mean Average Precision (MAP) on both Image-to-Text (I2T) and Text-to-Image (T2I) retrieval tasks. Note that all MAP scores are computed over the entire retrieval set (i.e., MAP@ALL). Additionally, precision–recall curves under the hash lookup protocol are employed to visually assess

the performance of CMH. To distinguish between different levels of redundant annotations, we define a redundant rate, which represents the ratio between the number of redundant noisy labels and the number of ground-truth labels in the entire annotations. Experiments were conducted with hash code lengths of 32, 64, and 128 bits, under redundant rates of 1.0, 1.5, 2.0, and 2.5. Additional details about the datasets are provided in Appendix Sec. B.

## 4.2 Implementation Details

In the proposed NACD, the image modality adopts the VGG19 model [45], pre-trained on ImageNet, as its CNN backbone. For text processing, the pre-trained Doc2Vec model [46] is employed as the backbone. For cross-modal shared representation learning, the image and text modalities employ three and two hidden layers, respectively. Each fully connected (FC) layer is succeeded by a ReLU activation layer, except for the final layer, which uses a tanh function. Each hidden layer contains 8,192 units, followed by an output layer of dimension $L$ representing the shared embedding space. The model is trained using the RMSprop optimizer [47], with an initial learning rate of $1e-5$ and a maximum of 100 epochs. The parameters $\delta$ and $\xi$ in Eq. (8) are set to 0.2 and 1.0, respectively. Additionally, we employ a batch size n of 128. The model is evaluated every 20 epochs, with the first 10 epochs serving as a warm-up phase during which the CM strategy and class-wise threshold update are disabled. The number of neighbors $C$ is set to 20 to ensure accurate neighborhood information. Our NACD is implemented using the PyTorch framework [48] and all experiments are carried out with 4 NVIDIA V100 GPUs.

## 4.3 Comparison with State-of-the-Arts

In this work, we compare our NACD against 11 state-of-the-art CMH methods, including five unsupervised methods: DJSRH [18], DGCPN [20], PIP [49], CIRH [21], and UCCH [22]; and six supervised methods: CMMQ [50], MIAN [27], LtCMH [28], DHRL [29], NRCH [30], and RSHNL [31]. The average MAP scores for the I2T and T2I tasks are reported in Table 1. Additionally, the experiment results with 8 and 16 bits can be found in Appendix Sec. C.1. Fig. 3 presents precision-recall curves on three datasets for a hash code length of 128 bits and a redundant rate of 2.5. Based on these results, we make the following observations:

- As the hash code length increases, the performance of almost all methods improves, since longer codes contain more discriminative information in the Hamming space.

- As the redundant rate increases, the performance of all supervised CMH methods deteriorates, since the progressively redundant noisy supervision misleads model training. In contrast, NACD maintains stable and superior performance by effectively extracting correct supervision from redundant annotations. Meanwhile, unsupervised CMH methods remain unaffected as they do not rely on label information.

- From Table 1, we can see that the proposed NACD consistently outperforms all competing methods across all settings. For instance, when the hash code length is 128 and the redundant rate is 2.5, NACD exceeds the second-best methods by 3.3%, 1.7%, and 2.6% on the Flickr, NUS, and COCO datasets, respectively.

- As illustrated in Fig. 3, the area under the precision–recall curves indicates that NACD consistently outperforms all other state-of-the-art methods in both I2T and T2I tasks, demonstrating its stable and superior performance.

## 4.4 Ablation Study

To verify the effectiveness of each component in NACD, we conducted extensive ablation studies on the NUS and COCO datasets with a hash code length of 128 bits across various redundant rates. We compared the full NACD with six ablated variants: (1) only $\mathcal{L}_{nacr}$; (2) only $\mathcal{L}_{crch}$; (3) NACD without the CM strategy; (4) NACD without the class-wise threshold $t_k$ fixed at 0.5; (5) NACD without $\mathcal{L}_{quant}$; (6) NACD without the warm-up phase; and (7) the full NACD. As shown in Table 2, NACR, CRCH, and $\mathcal{L}_{quant}$ all effectively enhance the performance of NACD. Additionally, the warm-up strategy, the CM strategy, and dynamic class-wise threshold updating are all crucial for achieving optimal model performance.

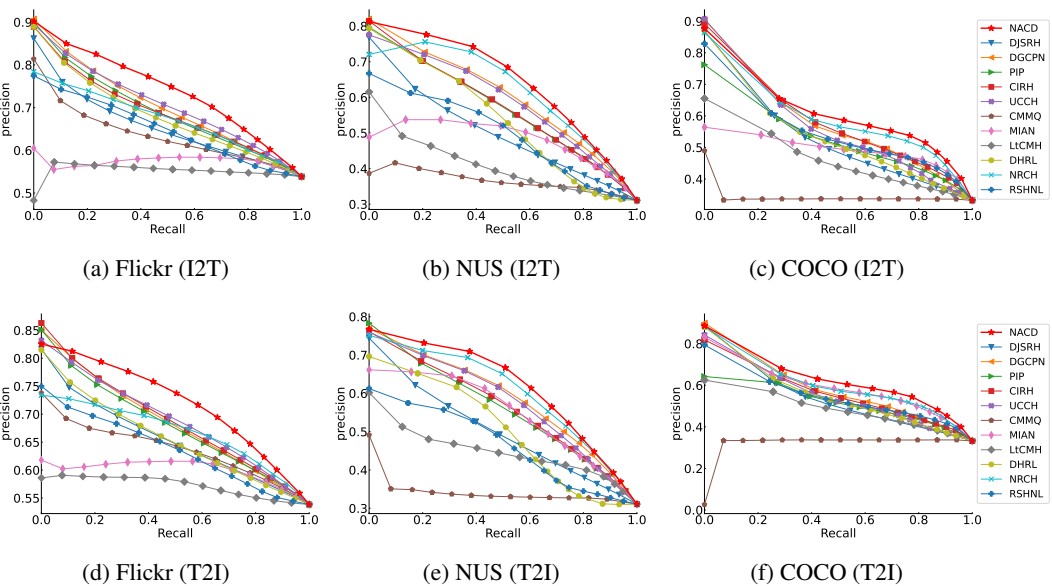

| (a) Flickr (I2T) | (b) NUS (I2T) | (c) COCO (I2T) |
| --- | --- | --- |

| (d) Flickr (T2I) | (e) NUS (T2I) | (f) COCO (T2I) |
| --- | --- | --- |

Figure 3: The precision-recall curves on three datasets. Note that the hash code length is 128bits and the redundant rate is 2.5.

## 4.5 Parameter Analysis

To evaluate the impact of the coefficient $\lambda$ in Eq. (3), as well as $\alpha$ and $\beta$ in Eq. (11), we conducted extensive experiments on the COCO dataset with a hash code length of 128 bits under various redundant rates. As shown in Fig. 4, $\lambda$ yields optimal results within the range of $[0.01, 0.04]$, demonstrating the effectiveness of NACR. Moreover, the model achieves superior performance when $\alpha$ lies within $[0.1, 0.5]$ and $\beta$ within $[0.5, 1.5]$, further validating the effectiveness of both CRCH and the quantization loss $\mathcal{L}_{quant}$. Additional parameter analyses on the Flick and NUS datasets are provided in Appendix Sec. C.2.

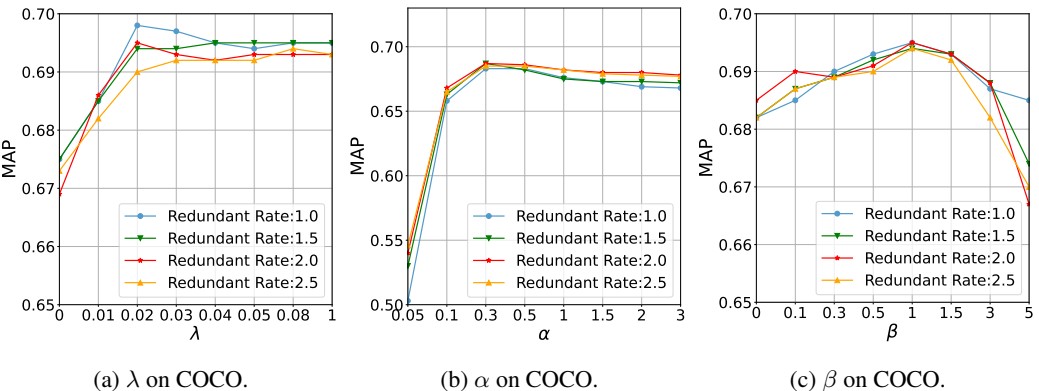

| (a) $\lambda$ on COCO. | (b) $\alpha$ on COCO. | (c) $\beta$ on COCO. |
| --- | --- | --- |

Figure 4: The performance of NACD in terms of average MAP scores versus different values of $\lambda$, $\alpha$, and $\beta$ on COCO dataset with 128 bits.

## 4.6 Model Analysis

**Robustness Analysis.** To intuitively demonstrate the robustness of NACD, we compared it with two of its variants and the NRCH method. Their average MAP scores on the COCO dataset were plotted under the settings of a hash code length of 128 bits and redundant rates of 2.0 and 2.5. Specifically, NACD-1 denotes the variant without the CM strategy, while NACD-2 represents the variant in which the class-wise threshold $t_k$ is fixed at 0.5, meaning that no dynamic threshold update is performed.

Table 1: The performance comparison in terms of average MAP scores (%) of I2T and T2I tasks under different redundant rates and various bit lengths on the MIRFlickr-25K(Flickr), NUS-WIDE(NUS), and MS-COCO(COCO) datasets. The highest and second highest MAP scores among all methods are shown in **bold** and underline respectively.

| Dataset | Method | Year | 1.0 | | | 1.5 | | | 2.0 | | | 2.5 | | |
|---|---|---|---|---|---|---|---|---|---|---|---|---|---|---|
| | | | 32bits | 64bits | 128bits | 32bits | 64bits | 128bits | 32bits | 64bits | 128bits | 32bits | 64bits | 128bits |
| Flickr | DJSRH | 2019 | 63.4 | 64.8 | 66.5 | 63.4 | 64.8 | 66.5 | 63.4 | 64.8 | 66.5 | 63.4 | 64.8 | 66.5 |
| | DGCPN | 2021 | 69.2 | 70.5 | 71.1 | 69.2 | 70.5 | 71.1 | 69.2 | 70.5 | 71.1 | 69.2 | 70.5 | 71.1 |
| | PIP | 2021 | 67.9 | 68.9 | 69.8 | 67.9 | 68.9 | 69.8 | 67.9 | 68.9 | 69.8 | 67.9 | 68.9 | 69.8 |
| | CIRH | 2022 | 68.6 | 69.4 | 70.1 | 68.6 | 69.4 | 70.1 | 68.6 | 69.4 | 70.1 | 68.6 | 69.4 | 70.1 |
| | UCCH | 2023 | 71.0 | 71.9 | 72.0 | 71.0 | 71.9 | 72.0 | 71.0 | 71.9 | 72.0 | 71.0 | 71.9 | 72.0 |
| | CMMQ | 2022 | 63.6 | 64.1 | 65.5 | 66.5 | 67.6 | 69.1 | 63.6 | 63.9 | 65.6 | 63.6 | 63.8 | 64.8 |
| | MIAN | 2023 | 71.1 | 71.8 | 73.0 | 67.6 | 68.4 | 68.0 | 61.5 | 62.8 | 63.2 | 61.6 | 61.3 | 60.2 |
| | LtCMH | 2023 | 63.2 | 64.3 | 65.3 | 60.0 | 59.8 | 60.6 | 57.9 | 57.8 | 58.5 | 56.9 | 56.4 | 57.6 |
| | DHRL | 2024 | 69.5 | 69.4 | 69.3 | 68.8 | 69.0 | 68.6 | 69.2 | 68.9 | 68.9 | 69.0 | 68.8 | 68.4 |
| | NRCH | 2024 | 72.9 | 74.5 | 74.4 | 67.9 | 70.9 | 69.5 | 63.0 | 63.6 | 65.1 | 57.7 | 58.2 | 63.9 |
| | RSHNL | 2025 | 71.8 | 72.2 | 72.5 | 69.2 | 69.5 | 70.5 | 66.5 | 67.5 | 67.9 | 64.4 | 65.0 | 66.0 |
| | **NACD** | **Ours** | **76.5** | **76.8** | **77.1** | **76.4** | **76.8** | **76.9** | **75.7** | **76.2** | **76.3** | **74.3** | **74.6** | **75.3** |
| NUS | DJSRH | 2019 | 46.7 | 49.5 | 52.1 | 46.7 | 49.5 | 52.1 | 46.7 | 49.5 | 52.1 | 46.7 | 49.5 | 52.1 |
| | DGCPN | 2021 | 60.2 | 62.4 | 64.0 | 60.2 | 62.4 | 64.0 | 60.2 | 62.4 | 64.0 | 60.2 | 62.4 | 64.0 |
| | PIP | 2021 | 57.3 | 59.1 | 59.9 | 57.3 | 59.1 | 59.9 | 57.3 | 59.1 | 59.9 | 57.3 | 59.1 | 59.9 |
| | CIRH | 2022 | 57.2 | 59.3 | 60.4 | 57.2 | 59.3 | 60.4 | 57.2 | 59.3 | 60.4 | 57.2 | 59.3 | 60.4 |
| | UCCH | 2023 | 62.3 | 63.4 | 63.9 | 62.3 | 63.4 | 63.9 | 62.3 | 63.4 | 63.9 | 62.3 | 63.4 | 63.9 |
| | CMMQ | 2022 | 57.7 | 57.7 | 58.3 | 52.5 | 52.5 | 52.7 | 44.8 | 44.2 | 44.1 | 36.9 | 36.4 | 36.8 |
| | MIAN | 2023 | 63.5 | 63.6 | 64.1 | 58.9 | 60.2 | 61.2 | 57.4 | 58.1 | 59.2 | 57.1 | 56.5 | 58.6 |
| | LtCMH | 2023 | 57.0 | 58.0 | 59.4 | 51.7 | 52.5 | 53.2 | 50.6 | 49.5 | 49.5 | 45.3 | 45.5 | 46.6 |
| | DHRL | 2024 | 61.0 | 61.0 | 60.8 | 60.3 | 60.2 | 58.9 | 58.7 | 59.2 | 57.8 | 57.7 | 58.7 | 57.4 |
| | NRCH | 2024 | 67.9 | 68.6 | 69.3 | 67.4 | 68.3 | 68.4 | 67.0 | 67.7 | 68.0 | 66.6 | 67.3 | 67.6 |
| | RSHNL | 2025 | 59.3 | 59.7 | 60.3 | 57.8 | 57.7 | 57.5 | 55.5 | 55.5 | 55.7 | 54.4 | 54.0 | 53.0 |
| | **NACD** | **Ours** | **68.2** | **69.4** | **70.3** | **68.1** | **69.4** | **70.2** | **67.9** | **69.0** | **69.6** | **67.4** | **68.7** | **69.3** |
| COCO | DJSRH | 2019 | 51.6 | 54.1 | 57.0 | 51.6 | 54.1 | 57.0 | 51.6 | 54.1 | 57.0 | 51.6 | 54.1 | 57.0 |
| | DGCPN | 2021 | 63.0 | 63.5 | 64.3 | 63.0 | 63.5 | 64.3 | 63.0 | 63.5 | 64.3 | 63.0 | 63.5 | 64.3 |
| | PIP | 2021 | 55.7 | 57.8 | 58.2 | 55.7 | 57.8 | 58.2 | 55.7 | 57.8 | 58.2 | 55.7 | 57.8 | 58.2 |
| | CIRH | 2022 | 63.0 | 63.5 | 64.2 | 63.0 | 63.5 | 64.2 | 63.0 | 63.5 | 64.2 | 63.0 | 63.5 | 64.2 |
| | UCCH | 2023 | 60.4 | 61.4 | 61.8 | 60.4 | 61.4 | 61.8 | 60.4 | 61.4 | 61.8 | 60.4 | 61.4 | 61.8 |
| | CMMQ | 2022 | 43.9 | 44.1 | 44.6 | 39.8 | 39.8 | 40.2 | 35.9 | 35.5 | 35.5 | 33.7 | 33.7 | 33.7 |
| | MIAN | 2023 | 61.0 | 63.8 | 64.8 | 60.8 | 62.9 | 63.6 | 59.2 | 60.2 | 60.5 | 57.3 | 59.4 | 58.7 |
| | LtCMH | 2023 | 58.1 | 60.5 | 62.1 | 56.3 | 56.3 | 59.6 | 54.7 | 56.3 | 58.9 | 52.3 | 54.2 | 55.3 |
| | DHRL | 2024 | 33.4 | 34.5 | 61.9 | 33.3 | 43.6 | 60.4 | 33.2 | 43.2 | 61.5 | 35.2 | 60.6 | 59.8 |
| | NRCH | 2024 | 65.9 | 67.3 | 67.8 | 65.6 | 66.9 | 67.3 | 64.9 | 66.8 | 67.1 | 63.2 | 65.3 | 66.2 |
| | RSHNL | 2025 | 60.7 | 60.3 | 60.2 | 61.1 | 61.5 | 61.7 | 59.9 | 59.9 | 61.7 | 60.7 | 59.1 | 60.3 |
| | **NACD** | **Ours** | **66.7** | **68.4** | **68.7** | **67.2** | **68.4** | **69.1** | **67.1** | **68.5** | **69.0** | **66.5** | **68.3** | **68.8** |

Table 2: The ablation study results on NUS and COCO datasets with 128 bits and across various redundant rates. The highest scores are presented in bold.

| Method | NUS | | | | COCO | | | |
|---|---|---|---|---|---|---|---|---|
| | 1.0 | 1.5 | 2.0 | 2.5 | 1.0 | 1.5 | 2.0 | 2.5 |
| only $\mathcal{L}_{nacr}$ | 34.3 | 33.5 | 33.3 | 36.0 | 33.3 | 33.3 | 33.3 | 33.3 |
| only $\mathcal{L}_{crch}$ | 63.3 | 63.0 | 63.1 | 63.2 | 64.9 | 64.9 | 65.6 | 65.7 |
| NACD w/o CM strategy | 69.2 | 69.1 | 69.2 | 69.0 | 67.2 | 66.9 | 67.1 | 67.1 |
| NACD with $t_k$ remains 0.5 | 70.0 | 69.5 | 69.3 | 68.9 | 67.9 | 68.3 | 68.3 | 68.0 |
| NACD w/o $\mathcal{L}_{quant}$ | 69.0 | 69.0 | 68.4 | 68.1 | 68.0 | 67.9 | 67.9 | 67.8 |
| NACD w/o warm-up | 69.3 | 69.2 | 68.6 | 68.4 | 68.0 | 68.1 | 68.2 | 68.1 |
| The full NACD | **70.3** | **70.2** | **70.2** | **69.6** | **68.7** | **69.1** | **69.0** | **68.8** |

As shown in Fig. 5(a,b): (1) Both NACD-1 and NACD-2 tend to overfit redundant noise, indicating that the CM strategy and class-wise threshold updating are crucial for enhancing the robustness of NACD. (2) Although NRCH exhibits certain robustness under redundant annotations, it still lags

significantly behind NACD. This gap widens as noise levels increase, further underscoring NACD's robustness in challenging scenarios.

**Disambiguation Analysis.** In Fig. 5(c), we analyze the model's ability to disambiguate redundant annotations by computing the average pseudo-label length for all sample pairs in the COCO dataset. Here, the pseudo-label length for the $j$-th sample pair is defined as the number of positive entries in $\hat{y}_j$, reflecting how many classes the model predicts as positive. As shown in the figure, when the redundant rate decreases, the average pseudo-label length gradually approaches the average number of ground-truth labels in the COCO dataset (2.76). This indicates that NACD can accurately recover the true label subset from redundant annotations, confirming its strong disambiguation capability.

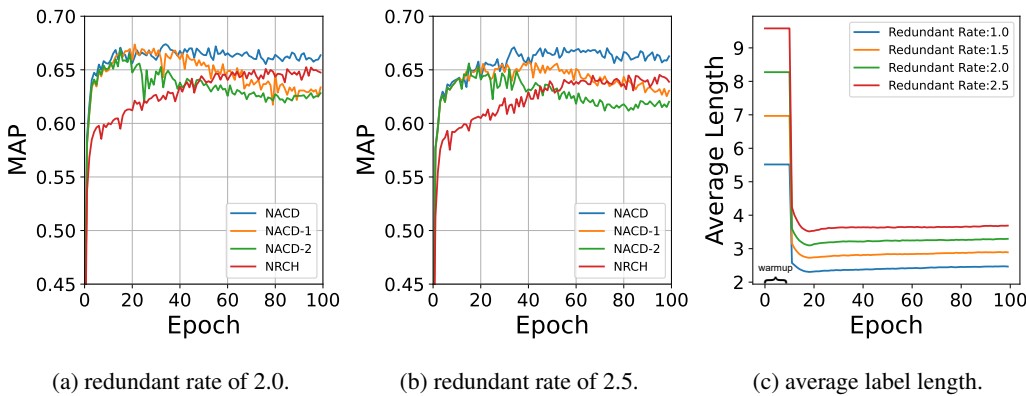

(a) redundant rate of 2.0.    (b) redundant rate of 2.5.    (c) average label length.

Figure 5: The robustness study and disambiguation study results with 32 bits on the COCO dataset.

## 5 Conclusion

In this paper, we propose a novel Neighbor-aware Contrastive Disambiguation (NACD) framework to address the challenge of redundant annotations in cross-modal hashing. NACD comprises two key modules: Neighbor-aware Confidence Reconstruction (NACR) and Class-aware Robust Contrastive Hashing (CRCH). Specifically, NACR reconstructs label confidence by aggregating information from cross-modal neighbors, thereby distinguishing true labels from ambiguous ones. Meanwhile, CRCH constructs reliable positive and negative pairs based on label confidence, substantially enhancing robustness under noisy supervision. Moreover, a quantization loss is incorporated to reduce the quantization error and enforce binary constraints on the learned hash representations. Extensive experiments on three large-scale multimodal benchmarks demonstrate that NACD consistently outperforms state-of-the-art approaches, showing strong robustness and stable performance for cross-modal hashing with redundant annotations.

**Limitation.** Although our proposed NACD demonstrates strong performance, there are still some limitations that need to be addressed. In this paper, the "redundant annotations" we study in cross-modal hashing do not account for the inherent similarity between labels. NACD may struggle when noisy labels are highly similar to the ground-truth, such as when the correct label is "car" and the noisy label is "truck". Additionally, we only conduct extensive experiments on image and text modalities to demonstrate NACD's effectiveness. In the future, additional modalities need to be considered to verify the generalization ability of NACD. We encourage further research to better understand and mitigate the limitations and risks of cross-modal hashing with redundant annotations.

## Acknowledgments

This work was supported by the National Natural Science Foundation of China (62306197, 62372315), Sichuan Science and Technology Planning Project (2025ZNSFSC1507, 2024YFG0007, 2024ZDZX0004, 2024NSFTD0049), China Postdoctoral Science Foundation (2021TQ0223, 2022M712236), Chengdu Science and Technology Project (2023-XT00-00004-GX), Postdoctoral Joint Training Program of Sichuan University (SCDXLHPY2307).

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

# Appendix

In the following sections, we provide additional details about the optimization process (Sec. A), the datasets (Sec. B), and the experimental results (Sec. C) of our proposed NACD.

## A    Optimization

To effectively learn under redundant annotation supervision, we design an end-to-end optimization strategy, as summarized in Algorithm 1. During the warm-up phase, we train the model without the CM strategy and the class-wise threshold update. Once the warm-up phase is complete, the stabilized model produces more reliable feature representations, which in turn provide stronger support for updating label confidences through neighborhood-based mechanisms. In addition, before backpropagation, we slightly update the label confidences of the current mini-batch based on the model's latest predictions to refine the supervision signals.

---

**Algorithm 1** Optimization Algorithm for NACD

---

**Require:** The redundant annotations training set $\mathcal{D}$, the code length $L$, the network $N = \{f_1(\cdot, \Theta_1), f_2(\cdot, \Theta_2)\}$, the learnable matrix $\mathbf{W_1}, \mathbf{W_2}$, the maximal epoch number $T_{\max}$, and the warm-up epoch number $T_{\mathrm{warm}}$;
1: Randomly initialize network parameters $\{\Theta_i, W_i\}_{i=1}^2$;
2: Use part of the multi-partial labels of each sample as the initial state of instance confidence, and set the threshold for each class to 0.5.
3: **for** $epoch = 1$ to $T_{\max}$ **do**
4:     **for** $\mathcal{D}_n$ in mini-batches sampled from $\mathcal{D}$ **do**
5:         Compute $L_{final}$ by Eq. (11);
6:         Update instance confidence by Eq. (1);
7:         Optimize network parameters through backpropagation;
8:     **end for**
9:     **if** $epoch > T_{\mathrm{warm}}$ **then**
10:         Select the neighbor update instance label confidence by Eq. (2);
11:         Update class-aware threshold by Eq. (5);
12:     **end if**
13: **end for**
**Ensure:** Network parameters $\{\Theta_i, W_i\}_{i=1}^2$.

---

## B    Datasets

To verify the effectiveness of our proposed NACD method in addressing the redundant annotations problem, we conduct experiments on the MIRFlickr-25k (Flickr) [42], NUS-WIDE (NUS) [43], and MS-COCO (COCO) [44] datasets. The details are as follows: MIRFlickr-25K [42] contains 25,000 image-text pairs, each belonging to one of 24 categories. In this work, we only select 20,015 pairs that have annotations. NUS-WIDE [43] is a multimodal dataset containing 81 concept categories. In this work, we only consider the subset of data from the most frequent 21 categories, which includes 190,421 image-text pairs. MS-COCO [44] encompasses a vast collection of 123,287 images distributed across 80 diverse categories. Each image is enriched with five detailed textual descriptions. After considering only labeled data, we ultimately select 122,218 image-text pairs.

Following dataset partition strategies adopted in prior works [22, 30], we have structured the datasets accordingly: For MIRFlickr-25K [42], we select 2,000 data points as the test (query) dataset. The remaining data points are used to form the retrieval (database) dataset. From this, we further identified a training subset comprising 10,000 data points. In the case of NUS-WIDE [43], the test (query) dataset is made up of 2,100 data points. The remaining data points form the retrieval (database) dataset. From this dataset, we select 10,500 data points for training. For MS-COCO [44], we extract 5,000 data points to be used for testing. The rest of the data points are pooled into the retrieval (database) dataset. From this, we set aside 10,000 data points specifically for training. Table 3 shows the specific data split information of these three multimodal datasets in our experiments.

Table 3: Data split and basic information for Flickr, NUS, and COCO in our experiments.

| Dataset | Test (query) | Database | Train | Average length of GTs | Classes |
|---------|-------------|----------|-------|----------------------|---------|
| Flickr | 2,000 | 18,015 | 10,000 | 3.78 | 24 |
| NUS | 2,100 | 188,321 | 10,500 | 2.09 | 21 |
| COCO | 5,000 | 117,218 | 10,000 | 2.76 | 80 |

## C Experimental Results

### C.1 Additional Comparative Experiments

Moreover, to further validate the effectiveness of our method in a more comprehensive manner, we conduct additional experiments using 8-bit and 16-bit hash codes. The results of these experiments are presented in Table 4, demonstrating the stable and competitive performance of our approach under various experimental settings.

### C.2 Additional Parameter Analysis

To analyze the impact of the coefficients $\lambda$, $\alpha$, and $\beta$ in Eq. (3) and Eq. (11), we conduct parameter analysis experiments on the Flickr and NUS datasets under different redundant rates. The results are illustrated in Fig. 6. Specifically, for the Flickr dataset, the model achieves its peak performance when $\lambda$ is within [0.1, 0.3], $\alpha$ within [0.1, 0.5], and $\beta$ within [0.5, 1]. This indicates that relatively balanced tuning of these parameters is crucial to optimize the model's performance on this dataset. Similarly, for the NUS dataset, the model exhibits optimal performance when $\lambda$ is within [0.05, 0.5], $\alpha$ within [0.3, 1], and $\beta$ within [0.5, 1]. These findings highlight the necessity of dataset-specific parameter tuning to maximize model performance.

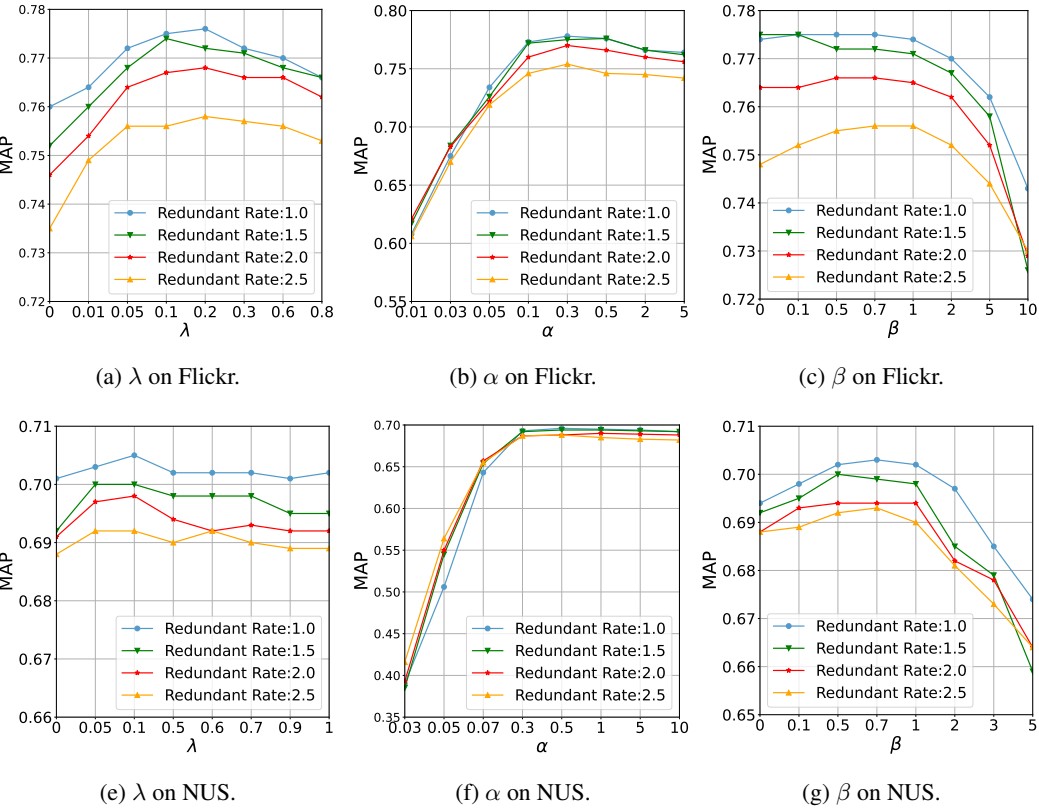

(a) $\lambda$ on Flickr.     (b) $\alpha$ on Flickr.     (c) $\beta$ on Flickr.

(e) $\lambda$ on NUS.     (f) $\alpha$ on NUS.     (g) $\beta$ on NUS.

Figure 6: The performance of NACD in terms of average MAP scores versus different values of $\lambda$, $\beta$ and $\alpha$ on the Flickr and NUS datasets using 128 bits.

Table 4: The performance comparison in terms of average MAP scores (%) of I2T and T2I tasks under different redundant rates and various bit lengths on the MIRFlickr-25K(Flickr), NUS-WIDE(NUS), and MS-COCO(COCO) datasets. The highest and second highest MAP scores among all methods are shown in **bold** and underline respectively.

| Dataset | Method | Year | 1.0 | | 1.5 | | 2.0 | | 2.5 | |
|---|---|---|---|---|---|---|---|---|---|---|
| | | | 8bits | 16bits | 8bits | 16bits | 8bits | 16bits | 8bits | 16bits |
| Flickr | DJSRH | 2019 | 61.3 | 61.4 | 61.3 | 61.4 | 61.3 | 61.4 | 61.3 | 61.4 |
| | DGCPN | 2021 | 69.0 | 67.9 | 69.0 | 67.9 | 69.0 | 67.9 | 69.0 | 67.9 |
| | PIP | 2021 | 66.9 | 67.4 | 66.9 | 67.4 | 66.9 | 67.4 | 66.9 | 67.4 |
| | CIRH | 2022 | 65.6 | 66.8 | 65.6 | 66.8 | 65.6 | 66.8 | 65.6 | 66.8 |
| | UCCH | 2023 | 67.5 | 70.1 | 67.5 | 70.1 | 67.5 | 70.1 | 67.5 | 70.1 |
| | CMMQ | 2022 | 58.1 | 60.5 | 61.9 | 62.7 | 56.5 | 61.8 | 56.5 | 61.8 |
| | MIAN | 2023 | 65.5 | 68.6 | 62.1 | 64.1 | 59.7 | 61.7 | 58.9 | 59.9 |
| | LtCMH | 2023 | 60.7 | 61.4 | 58.7 | 57.8 | 55.6 | 54.9 | 56.2 | 55.0 |
| | DHRL | 2024 | 67.1 | 68.7 | 65.7 | 68.2 | 65.3 | 67.6 | 65.3 | 66.5 |
| | NRCH | 2024 | 67.0 | 71.0 | 53.9 | 58.5 | 53.9 | 57.9 | 53.9 | 56.9 |
| | RSHNL | 2025 | 70.3 | 71.5 | 68.8 | 71.0 | 68.7 | 70.2 | 68.9 | 70.9 |
| | **NACD** | **Ours** | **73.9** | **75.9** | **73.6** | **75.0** | **72.5** | **74.8** | **70.2** | **73.3** |
| NUS | DJSRH | 2019 | 42.8 | 42.9 | 42.8 | 42.9 | 42.8 | 42.9 | 42.8 | 42.9 |
| | DGCPN | 2021 | 55.2 | 58.6 | 55.2 | 58.6 | 55.2 | 58.6 | 55.2 | 58.6 |
| | PIP | 2021 | 55.7 | 56.2 | 55.7 | 56.2 | 55.7 | 56.2 | 55.7 | 56.2 |
| | CIRH | 2022 | 54.6 | 55.5 | 54.6 | 55.5 | 54.6 | 55.5 | 54.6 | 55.5 |
| | UCCH | 2023 | 57.4 | 60.1 | 57.4 | 60.1 | 57.4 | 60.1 | 57.4 | 60.1 |
| | CMMQ | 2022 | 56.4 | 57.4 | 52.1 | 52.3 | 46.2 | 45.5 | 38.5 | 37.6 |
| | MIAN | 2023 | 58.9 | 62.1 | 54.8 | 55.2 | 54.2 | 54.4 | 50.0 | 54.2 |
| | LtCMH | 2023 | 52.2 | 53.0 | 36.9 | 46.1 | 38.6 | 45.3 | 38.0 | 44.1 |
| | DHRL | 2024 | 58.3 | 59.3 | 54.0 | 58.9 | 54.0 | 57.3 | 52.6 | 56.8 |
| | NRCH | 2024 | **63.7** | **65.9** | **61.8** | 65.4 | 61.1 | 64.9 | 59.2 | 64.6 |
| | RSHNL | 2025 | 58.2 | 61.1 | 53.9 | 57.6 | 54.5 | 56.0 | 50.9 | 55.0 |
| | **NACD** | **Ours** | 62.9 | 65.0 | 61.3 | **66.1** | **61.3** | **65.8** | **62.1** | **65.2** |
| COCO | DJSRH | 2019 | 41.9 | 47.6 | 41.9 | 47.6 | 41.9 | 47.6 | 41.9 | 47.6 |
| | DGCPN | 2021 | 56.6 | 61.1 | 56.6 | 61.1 | 56.6 | 61.1 | 56.6 | 61.1 |
| | PIP | 2021 | 46.2 | 52.6 | 46.2 | 52.6 | 46.2 | 52.6 | 46.2 | 52.6 |
| | CIRH | 2022 | 54.2 | 59.6 | 54.2 | 59.6 | 54.2 | 59.6 | 54.2 | 59.6 |
| | UCCH | 2023 | 55.3 | 57.0 | 55.3 | 57.0 | 55.3 | 57.0 | 55.3 | 57.0 |
| | CMMQ | 2022 | 45.4 | 44.2 | 41.5 | 39.8 | 36.1 | 35.6 | 33.8 | 33.7 |
| | MIAN | 2023 | 53.9 | 58.5 | 55.0 | 58.4 | 53.0 | 57.4 | 52.4 | 56.2 |
| | LtCMH | 2023 | 52.3 | 54.7 | 51.2 | 51.2 | 50.0 | 51.5 | 40.0 | 49.6 |
| | DHRL | 2024 | 34.4 | 33.4 | 33.4 | 33.8 | 33.4 | 33.8 | 33.4 | 42.1 |
| | NRCH | 2024 | 59.5 | 64.2 | 58.7 | 63.3 | 57.5 | 62.0 | 56.2 | 61.0 |
| | RSHNL | 2025 | 56.6 | 59.3 | 55.4 | 59.7 | 57.6 | 61.3 | 55.2 | 59.3 |
| | **NACD** | **Ours** | **62.1** | **65.5** | **61.7** | **65.4** | **62.3** | **65.6** | **60.5** | **65.8** |

