# OpenReview forum: "Neighbor-aware Contrastive Disambiguation for Cross-Modal Hashing with Redundant Annotations"
_NeurIPS.cc/2025/Conference — NeurIPS 2025 spotlight_

### Official Review · Reviewer_ETEA · 2025-06-25

**Clarity:** 3
**Significance:** 3
**Originality:** 3
**Rating:** 5
**Confidence:** 5

**Summary:**

The paper proposes a novel Neighbor-aware Contrastive Disambiguation (NACD) approach, which is the first work to focus on the scenario of redundant annotations in the cross-modal hashing task. This work primarily involves two key innovative modules: a Neighbor-aware Confidence Reconstruction (NACR) which refines label confidence by aggregating information from cross-modal neighbors to achieve label disambiguation, and a Class-aware Robust Contrastive Hashing (CRCH) module for robustly reducing modality gaps under redundant supervision. Additionally, this work further enhances model performance by introducing an innovative quantization loss that enforces binary constraints on the learned hash representations. Importantly, the experiments in this paper are extremely comprehensive and meticulous, not only demonstrating the advanced performance of NACD and the effectiveness of each module, but also providing additional evidence for the robustness of NACD and the effectiveness of label disambiguation through more experiments.

**Questions:**

- Referring to Eq (2), how should the number of neighbors and the computation be balanced?

- The Redundant Annotations in this paper assume all correct labels are annotated. How to consider scenarios where some true labels are missed?

**Ethical Concerns:**

["NO or VERY MINOR ethics concerns only"]

**Final Justification:**

The authors have addressed all my concerns, and taking into account the feedback from other reviewers, I recommend that this work be accepted.

**Limitations:**

Yes

**Quality:**

4

**Strengths And Weaknesses:**

Strengths:

- The problem explored in this paper---cross-modal hashing with redundant annotations---is very practical and interesting.

- The proposed method is generally very intuitive and I think it will be effective.

- The experiments in this paper are extremely comprehensive and meticulous.

Weaknesses:

- Why are the redundant rates in the paper set to 1.0, 1.5, 2.0, and 2.5? Are excessively high redundant rates not realistic in actual annotation scenarios?

- The role of the warm-up strategy needs to be clarified.

---

> ### Author Rebuttal · Authors · 2025-07-31
>
> Thank you for your valuable comment. Below are the responses to your comments.
>
> **Q1.** Why are the redundant rates in the paper set to 1.0, 1.5, 2.0, and 2.5? Are excessively high redundant rates not realistic in actual annotation scenarios?
>
> **A1.** Thank you for the insightful question. The redundant annotations setting is inspired by the real-world annotation practice of **"trying not to miss any ground truth"** — annotators tend to include more potentially relevant labels, especially for uncertain or difficult samples, to avoid missing true labels. Thus the redundant rates of 1.0, 1.5, 2.0, and 2.5 in our paper are chosen based on both practical annotation behavior and alignment with prior works.
>
> (1) we agree that excessively high redundant rates are not realistic in practical annotation, as **human annotators typically do not assign an overwhelming number of labels per instance**. To reflect this, our selected range (1.0–2.5) intentionally avoids extreme noise and instead focuses on realistic and representative levels of annotation redundancy.
>
> (2) our choice is **consistent with previous work[1]**, which adopt the same or similar redundant rate settings to evaluate robustness in partial multi-label learning frameworks. This consistency helps ensure fair comparison and benchmarking.
>
> In summary, the selected redundant rates provide a balanced coverage from mild to relatively challenging noise conditions, reflecting real-world tendencies while maintaining comparability with prior work.
>
> **References**
>
> [1] Hang J Y, Zhang M L. Partial multi-label learning with probabilistic graphical disambiguation[J]. Advances in Neural Information Processing Systems, 2023, 36: 1339-1351.
>
> **Q2.** The role of the warm-up strategy needs to be clarified.
>
> **A2.** Thank you for your insightful comment regarding the warm-up strategy. Our neighbor-aware confidence reconstruction relies on cross-modal neighborhood features. However, **during the warm-up phase, the model is not yet sufficiently trained**, and the output features—both across modalities and within the same modality—are still unstable. This instability can lead to inaccurate neighborhood identification, which in turn compromises the reliability of the reconstructed label confidence.
>
> To mitigate this issue, we **disable the neighbor-aware confidence reconstruction module during the warm-up stage and retain only the contrastive learning objective**. This allows the model to focus on reducing cross-modal discrepancies and stabilizing feature representations. Once the features become more reliable, we reintroduce the confidence reconstruction to provide a more accurate estimation of sample-level label confidence.
>
> As shown in the algorithm's pseudocode, we resume both the instance label confidence update and the computation of class-wise thresholds after the warm-up phase. We will improve the introduction of the role of the warm-up strategy in the next version of the paper.
>
> **Q3.** Referring to Eq (2), how should the number of neighbors and the computation be balanced?
>
> **A3.** Thank you for the insightful question. It is true that a small number of neighbors may lead to unreliable label estimation, while a large number of neighbors increases noise and computational cost. As a result, I followed previous conventions and set the number of neighbors to 20, which provides sufficient semantic context from relevant samples while maintaining tractable overhead. In the revised version of the paper, we will include a more detailed explanation regarding the choice of the number of neighbors.
>
> **Q4.** How to consider scenarios where some true labels are missed?
>
> **A4.** Thank you for the valuable question. Our current setting follows the standard Partial Multi-Label Learning (PML) framework, where the candidate label set (also referred to as redundant annotations) includes all ground-truth labels, along with a certain number of incorrect labels. This assumption reflects the common annotation practice of **"trying not to miss any ground truth"**, where annotators tend to include potentially relevant labels to avoid missing any true ones.
>
> However, we acknowledge that in real-world applications, missing true labels is also possible, leading to more challenging scenarios. Addressing such cases would require more advanced strategies. As illustrated in Figure 4(a)(b), the proposed NACD model can achieve robust retrieval performance under the interference of redundant noise labels. Moreover, Figure 4(c) intuitively demonstrates that NACD can effectively **extract the true labels from redundant annotations** progressively through training.
>
> We consider this direction highly meaningful and complementary to our current work. In future extensions, we plan to explore the scenarios where some true labels are missed, to better align with practical annotation noise.

---

> > ### Comment · Reviewer_ETEA · 2025-08-05
> >
> > Thank you for the thorough and well-organized rebuttal. The authors have addressed my concerns to my satisfaction. I maintain my original recommendation to accept the paper.

---

> > > ### Author Response · Authors · 2025-08-09
> > >
> > > We sincerely thank you for your insightful feedback, which has contributed greatly to refining our work.

---

### Official Review · Reviewer_2t6S · 2025-06-29

**Clarity:** 3
**Significance:** 3
**Originality:** 3
**Rating:** 5
**Confidence:** 5

**Summary:**

This paper proposes a new task called Cross-Modal Hashing with Redundant Annotations, which holds significant research value. The authors introduce the Neighbor-aware Contrastive Disambiguation (NACD) method to tackle the challenge of this new task: robust learning under redundant supervision. NACD incorporates two key components. The first, Neighbor-aware Confidence Reconstruction (NACR), refines label confidence by aggregating information from cross-modal neighbors to distinguish true labels from redundant noisy ones. The second, Class-aware Robust Contrastive Hashing (CRCH), constructs reliable positive and negative pairs based on label confidence scores, which significantly enhances robustness against noisy supervision. In addition to these components, this work also designs a warm-up strategy and a quantization loss to further boost retrieval performance. The paper is well-structured and easy to understand, effectively elucidating the research motivations. It is believed that this paper will bring new research ideas to the traditional task of cross-modal hashing.

**Questions:**

a) The principle of the quantization loss should be claried.

b) Please evaluate and explain the robustness of CRCH.

**Ethical Concerns:**

["NO or VERY MINOR ethics concerns only"]

**Final Justification:**

Thank you for the authors' responses. In the rebuttal, the autohrs explained the motivation and principle of the quantization loss, and robustness against redundant annotations. I raise my score to Accept.

**Limitations:**

Yes.

**Paper Formatting Concerns:**

NA.

**Quality:**

3

**Strengths And Weaknesses:**

Strengths:

a) Redundant Annotations is a labeling issue that is worth studying and has practical significance.

b) The empirical results show that NACD does help increase the performance for a dataset with Redundant Annotations.

c) The paper is highly persuasive in elaborating the motivations and highly innovative in NACD.

Weaknesses:

a) In the Introduction section, it is not quite clear what the differences and advantages are between Redundant Annotations and noise label annotations.

b) In section 4.5, how to understand the robustness demonstrated in Figure 4?

c) The font size in Figure 2 should be appropriately increased.

---

> ### Author Rebuttal · Authors · 2025-07-30
>
> Thank you for your valuable comment. Below are the responses to your comments.
>
> **Q1.** The differences and advantages are between redundant annotations and noisy label annotations.
>
> **A1.** Thank you for your comment. To the best of our knowledge, redundant annotations represent a novel paradigm in the field of cross-modal hashing that has not yet been explored. It differs from the conventional noisy label learning paradigm in the following aspects:
>
> (1) **In terms of label construction:** redundant annotations retain all original ground-truth labels and augment them with additional noisy labels to form a candidate label set. In contrast, noisy labels typically involve replacing some correct labels with incorrect ones.
>
> (2) **In terms of motivation:** The redundant annotations setting is inspired by the real-world annotation practice of **"trying not to miss any ground truth"** — annotators tend to include more potentially relevant labels, especially for uncertain or difficult samples, to avoid missing true labels. On the other hand, the noisy labels setting simulates annotation errors or misunderstandings, where label noise arises from subjective mistakes or system-level inaccuracies.
>
> (3) **In terms of learning challenges:** redundant annotations primarily focus on disambiguating the ground-truth labels from a larger candidate label set. In contrast, noisy label learning emphasizes identifying and handling incorrectly labeled samples.
> Therefore, redundant annotations better reflect practical annotation scenarios, where annotators prefer to include more candidate labels to ensure that the ground truth are not omitted — especially for ambiguous or challenging instances.
>
> Moreover, redundant annotations can retain as much correct label information as possible, which enables the model to fully leverage semantic supervision during training. This facilitates better modeling of cross-modal semantic correlations, thereby enhancing the discriminative power of the hash codes and improving retrieval performance. In the final version of the paper, we will provide a detailed explanation of the differences and advantages between redundant annotations and noisy label annotations in the introduction section.
>
> **Q2.** Explain the robustness demonstrated in Figure 4.
>
> **A2.** Thank you for the insightful question. Figure 4(a)(b) without the class-wise threshold mechanism, the NACD variant attains its highest MAP around 40 epochs before declining, indicating overfitting during training. Conversely, the complete NACD model either improves or maintains stable performance over 100 epochs, ultimately delivering superior results. Moreover, Figure 4(c) intuitively demonstrates that NACD can effectively extract the true labels from redundant annotations progressively through training. We will provide a detailed explanation in the final version of the robustness demonstrated in Figure 4.
>
> **Q3.** The font size in Figure 2 should be appropriately increased.
>
> **A3.** Thanks for your valuable suggestion. We will increase the font size in the figures and further enhance their appearance in the next version of the paper.
>
> **Q4.** The principle of the quantization loss.
>
> **Q4.** Thank you for your valuable comment. In cross-modal hashing, compact binary hash codes are employed to compute Hamming distances for efficient retrieval. However, during the training phase, the sign function—which is required for strict binarization—is non-differentiable and thus **unsuitable for gradient-based optimization**. To address this, continuous approximations such as the hyperbolic tangent (tanh) function are commonly adopted.
>
> As a result, the model generates real-valued features in the range of [-1, 1], rather than ideal binary codes. When these continuous features are eventually binarized using the sign function during inference, additional quantization error is introduced—leading to information loss that is not accounted for during training.
>
> To bridge this gap, a quantization loss is introduced as a regularization term. This loss encourages the real-valued outputs to remain close to their corresponding binary codes throughout training, thereby reducing the discrepancy between the learned features and the final hash codes. By aligning the optimization objective with the discrete nature of Hamming retrieval, quantization loss effectively **mitigates post-training information degradation** and improves overall retrieval performance.
>
> **Q5.** Evaluate and explain the robustness of CRCH.
>
> **A5.** Thank you for your insightful questions. CRCH leverages the reconstructed label confidences to establish a threshold for each category, enabling the generation of more reliable pseudo-labels. This facilitates the construction of more accurate positive and negative pairs for contrastive learning.
>
> In terms of contrastive learning, we adopt a label-similarity guided loss function for positive pairs, which effectively utilizes the improved label quality brought by class-wise thresholds. For negative pairs, we **further distinguish between hard negatives and easy negatives**, allowing for more targeted and effective learning.
>
> As illustrated in Figure 4(a)(b) of the paper, when the class-wise threshold mechanism is removed, the NACD variant reaches its peak MAP at around 40 epochs and then begins to decline, indicating overfitting during training. In contrast, the full version of NACD continues to improve or **remains stable throughout 100 epochs**, ultimately achieving better performance. We will provide more elaboration on the robustness of CRCH in the final version.

---

> > ### Comment · Reviewer_2t6S · 2025-08-06
> >
> > Thank you for the authors' responses. In the rebuttal, the autohrs explained the motivation and principle of the quantization loss, and robustness against redundant annotations. I raise my score to Accept.

---

> > > ### Author Response · Authors · 2025-08-09
> > >
> > > We deeply appreciate your careful review and insightful observations, which guided us in refining the paper.

---

### Official Review · Reviewer_pMFp · 2025-06-30

**Clarity:** 3
**Significance:** 3
**Originality:** 3
**Rating:** 5
**Confidence:** 4

**Summary:**

Unlike previous weakly supervised learning for cross-modal hashing, this paper, starting from the perspective of practical annotation, innovatively proposes a new weakly supervised learning paradigm called Redundant Annotations (RA). To achieve robust retrieval performance under redundant supervision, this paper specifically proposes the Neighbor-aware Contrastive Disambiguation (NACD) method, which includes the NACR and CRCH modules. Experiments have proven the effectiveness of both NACR and CRCH. Overall, this paper sets a new standard for the new problem of Cross-Modal Hashing with Redundant Annotations and is a solid and noteworthy piece of excellent work.

**Questions:**

Please refer to the weaknesses.

**Ethical Concerns:**

["NO or VERY MINOR ethics concerns only"]

**Final Justification:**

Thank you for the author's response. My problems have been properly resolved. I have decided to keep my score.

**Limitations:**

Yes

**Quality:**

3

**Strengths And Weaknesses:**

S1. The redundant annotation setting reveals a new weakly supervised learning paradigm in the cross-modal hashing task.
S2. The NACD method designed in the paper effectively addresses the problem of Redundant Annotations and achieves excellent retrieval performance.
S3. The experiments illustrate the effectiveness and efficiency of the proposed method.

W1. In section 3.3, further explanation is needed as to why the Intersection over Union (IoU) between pseudo-labels is used to compute the label similarity matrix.
W2. It needs to be explained why the CRCH module can achieve robust contrastive learning.
W3. Figure 4 (c) is not detailedly introduced.

---

> ### Author Rebuttal · Authors · 2025-07-30
>
> Thank you for your valuable comment. Below are the responses to your comments.
>
> **Q1.** In section 3.3, further explanation is needed as to why the Intersection over Union (IoU) between pseudo-labels is used to compute the label similarity matrix.
>
> **A1.** Thanks for your valuable comment. When constructing contrastive sample pairs, we treat pairs with no overlapping pseudo-labels as negative pairs. The remaining pairs, which share partially or fully overlapping labels, are considered positive pairs. Since these positive pairs exhibit **varying degrees of semantic similarity**, a soft metric is required to effectively capture their nuanced differences.
>
> To this end, we adopt the Intersection over Union (IoU) to measure label similarity. IoU focuses on the overall structure of the label sets and is inherently robust to the sparsity of multi-label annotations. Unlike simple vector-based similarity metrics, IoU is not affected by the absolute number of labels, thereby **eliminating bias caused by varying label lengths**. Moreover, it is less sensitive to the absence or inclusion of individual labels, leading to more stable and consistent similarity estimations across samples.
>
> **Q2.** It needs to be explained why the CRCH module can achieve robust contrastive learning.
>
> **A2.** Thank you for your valuable suggestions. To enable robust contrastive learning, CRCH relies on high-quality labels to construct meaningful positive and negative sample pairs. However, applying a fixed threshold across all categories fails to account for the variability in confidence distributions, which arise from differences in class difficulty, occurrence frequency, and semantic ambiguity.
>
> To address this, CRCH introduces a class-wise dynamic thresholding mechanism, which adaptively adjusts the threshold for each category based on the statistical properties of its reliable samples. This allows the model to obtain more reliable pseudo-labels while effectively filtering out noisy ones in a class-specific manner. As training progresses and confidence reconstruction becomes more accurate, the thresholds are dynamically updated to reflect the evolving label distributions.
>
> In the contrastive learning process, we adopt a label-similarity-guided loss function for positive pairs, which fully leverages the improved label quality resulting from class-wise thresholding. For negative pairs, we further **distinguish between hard and easy negatives**, enabling more targeted optimization and enhancing the discriminative power of the learned hash codes. We will provide a detailed explanation in the final version of the paper regarding why the CRCH module can achieve robust contrastive learning.
>
> **Q3.** Figure 4 (c) is not detailedly introduced.
>
> **A3.** Thank you for the insightful question. As shown in Figure 4, we present the variation in the average number of labels per sample on the MS-COCO dataset during training. The ground-truth average label set length in MS-COCO is 2.76. Under four different redundant ratios of 1.0, 1.5, 2.0, and 2.5, the initial average label set lengths of the noisy training set are 5.52, 6.9, 8.28, and 9.66, respectively.
>
> During the warm-up phase, since both the label confidence reconstruction and class-wise threshold updating are disabled, the average label set length remains unchanged. After the warm-up phase, a category is regarded as a pseudo-label only when its confidence exceeds the corresponding class-wise threshold. Under the joint effect of confidence reconstruction and adaptive thresholding, the average pseudo-label set length progressively approaches the true label set length.
>
> Moreover, we observe that the lower the redundancy ratio, the closer the final label set length is to the ground-truth average label set length. Even under higher redundancy settings, the number of retained labels per sample eventually becomes significantly smaller than the initial noisy label count. This demonstrates the effectiveness of our method in suppressing label noise and recovering clean supervision signals. We will improve the explanation of Figure 4(c) in the revised version of the paper.

---

> > ### Comment · Reviewer_pMFp · 2025-08-05
> >
> > Thank you for the author's response. My problems have been properly resolved. I have decided to keep my score.

---

> > > ### Author Response · Authors · 2025-08-09
> > >
> > > Thank you for your careful consideration and insightful comments, which greatly enhanced the quality of this work.

---

### Official Review · Reviewer_agwg · 2025-07-02

**Clarity:** 3
**Significance:** 4
**Originality:** 3
**Rating:** 4
**Confidence:** 4

**Summary:**

This paper innovatively investigate the redundant annotation problem in cross-modal hashing, which is of great research significance. The paper proposes a method termed Neighbor-aware Contrastive Disambiguation (NACD) to address this novel problem, which ensures robust learning under redundant supervision. Both the Neighbor-aware Confidence Reconstruction (NACR) and the Class-aware Robust Contrastive Hashing (CRCH) modules in NACD are highly innovative and achieve retrieval performance that far exceeds state-of-the-art methods. The comprehensive experiments well demonstrate the authors' viewpoints and motivations, endowing the paper with high research value.

**Questions:**

Please see the weakness and response each comment. Besides, Additionally, the authors should also answer the following question:
NACR heavily relies on the label information of neighbor sample pairs. Under annotation conditions with a high redundant rate, how can the performance of NACR be guaranteed?

**Ethical Concerns:**

["NO or VERY MINOR ethics concerns only"]

**Final Justification:**

The authors’ response has addressed my comments well. I will maintain my original score.

**Limitations:**

Yes

**Quality:**

3

**Strengths And Weaknesses:**

Strengths:

1)This paper is the first to study the redundant annotation problem in cross-modal hashing.

2)The proposed NACD achieves SOTA performance, especially demonstrating robust learning under redundant supervision.

3)The experiments in this paper are not only abundant but also well demonstrate the authors' viewpoints and motivations, endowing the paper with high research value.

4)The writing is straightforward, clear, and easy to understand.


Weaknesses:

1)The label connections between the anchor and neighbor sample pairs in Figure 1 are not well explained. These connections should be elaborated when introducing the NACR module based on Figure 1.

2)Further explanation is needed on why the dynamic class threshold can achieve robust contrastive learning. Compared with previous methods that use fixed thresholds, what are the advantages of the class-wise threshold updating strategy in CRCH?

3)In the section explaining the experimental results, there is no explanation as to why noise label learning methods such as DHRL, NRCH, and RSHNL are not able to effectively deal with Redundant Annotations.

---

> ### Author Rebuttal · Authors · 2025-07-31
>
> Thank you for your valuable comment. Below are the responses to your comments.
>
> **Q1.** The label connections between the anchor and neighbor sample pairs in Figure 1.
>
> **A1.** Thanks for your comment. In this paper, we investigate the task of cross-modal hashing between image and text modalities. Each image-text pair is treated as an anchor, and the similarity between the anchor and other sample pairs is computed as the average cosine similarity of their respective image and text components. Motivated by the assumption that **semantically similar samples tend to share common labels**, the top 20 most similar neighbors are selected based on this measure, and their label confidence is incorporated into the NACR module.
>
> Figure 1 illustrates an example of an anchor and one of its neighbors. The anchor has a redundant annotation set of **{Sea, Car, Mountain, Beach, Plant, Cloud, Dog, Human}**, among which **{Sea, Beach, Plant, Cloud}** are the true labels. The neighbor has a redundant annotation set of **{Sea, Horse, Airplane, Boat, Plant, Bridge, Building, Human}**, with **{Sea, Boat, Plant, Building}** being the true labels. While each sample pair possesses its own ground-truth label set, they exhibit partial overlap in true labels as **{Sea, Plant}**— a property that reflects the underlying semantic structure in feature space and motivates the design of the NACR mechanism.
>
> We will further clarify the label connections between the anchor and its neighbors in Figure 1 and elaborate on their role when introducing the NACR module in the revised version of the paper.
>
> **Q2.** Further explanation is needed on why the dynamic class threshold can achieve robust contrastive learning. Compared with previous methods that use fixed thresholds, what are the advantages of the class-wise threshold updating strategy in CRCH?
>
> **A2.** Thanks for your valuable question. Contrastive learning methods heavily rely on high-quality labels to construct effective positive and negative sample pairs. However, due to differences in class difficulty, occurrence frequency, and semantic ambiguity, the confidence distributions of pseudo-labels often vary significantly across categories. If the threshold is too loose for high-confidence classes, it may introduce noisy pseudo-labels; on the other hand, if it is too strict, it may incorrectly filter out truly relevant labels. Therefore, applying a dynamic threshold for each class is crucial for obtaining high-quality pseudo-labels. The class-wise dynamic threshold mechanism addresses this issue by aligning with the unique uncertainty characteristics of each category. It enables the retention of confident pseudo-labels while effectively filtering out noisy ones.
>
> In CRCH, class-wise thresholds are computed based on the statistical properties of reliable intra-class samples. These statistics are inherently robust to noise, allowing the model to suppress the influence of low-quality labels. More importantly, the thresholds are adaptively updated in response to the evolving confidence distribution. As the model improves and reconstructs more accurate label confidences, the dynamic thresholding mechanism evolves accordingly. This progressive refinement avoids the cumulative bias commonly observed in static thresholding methods, ensuring a continual improvement in pseudo-label quality throughout training. The results of the ablation study (Table 2) also well demonstrate the effectiveness of the class-wise threshold updating strategy.
>
> In the final version of the paper, we will further elaborate on the principles and advantages of the class-wise threshold updating strategy in the Methods section.
>
> **Q3.** In the section explaining the experimental results, there is no explanation as to why noise label learning methods such as DHRL, NRCH, and RSHNL are not able to effectively deal with Redundant Annotations.
>
> **A3.** Thank you for the insightful question. Methods like DHRL, NRCH, and RSHNL are designed for noisy label scenarios where only a subset of training samples are affected by label noise. These approaches typically aim to **identify and correct mislabeled samples** while assuming that some of the data remains clean. However, in the redundant annotations setting, every sample is associated with a candidate label set that includes both true and redundant labels. This means **the label information for every sample pair is ambiguous**, and the challenge becomes one of label disambiguation, not just noise filtering. Since these methods lack explicit mechanisms to separate true labels from redundant ones at the per-sample level, they are less effective under this more difficult setting, as demonstrated in our experimental results.
>
> In contrast, our method explicitly tackles this challenge through two key components: Neighbor-aware Confidence Reconstruction (NACR) and Class-aware Robust Contrastive Hashing (CRCH). NACR aggregates label confidence from semantically similar cross-modal neighbors to refine the confidence scores for each candidate label, enabling more accurate identification of true labels. Meanwhile, CRCH constructs contrastive pairs based on dynamically updated class-wise confidence thresholds, which helps reduce erroneous associations caused by noisy labels. Together, these modules allow our framework to effectively disambiguate labels and maintain robust performance, even under high levels of redundant supervision.
>
> In the final version of the paper, we will provide a more detailed explanation in the experiments section regarding the reasons for the poor retrieval performance of noisy label learning methods such as DHRL, NRCH, and RSHNL.
>
> **Q4.** NACR heavily relies on the label information of neighbor sample pairs. Under annotation conditions with a high redundant rate, how can the performance of NACR be guaranteed?
>
> **A4.** Thanks for your valuable comment. While NACR relies on the label information of neighbor sample pairs, it is robust under high redundant annotation rates. This is achieved through two core strategies: (1) cross-modal neighbor aggregation, which combines label confidence from both image and text modalities to reduce modality-specific noise; and (2) similarity-weighted fusion, which ensures that only semantically relevant neighbors have strong influence. These mechanisms help filter out noisy signals and reinforce true semantic patterns. As demonstrated in our main experiments, ablation studies, and robustness analysis (e.g., Table 1 , Table 2, and Fig. 4), NACR consistently maintains high performance even when the redundant rate reaches 2.5, demonstrating its robustness against noisy supervision.

---

> > ### Comment · Reviewer_agwg · 2025-08-09
> >
> > The authors have provided comprehensive responses to my initial concerns, particularly regarding the label connections in Figure 1, the dynamic threshold mechanism, and NACR's robustness. Their explanations, supported by additional experimental results, are convincing and effectively address the key issues raised. Overall, the paper presents valuable contributions, and the authors’ clarifications have reinforced my confidence in the work. I therefore maintain my original rating.

---

> > > ### Author Response · Authors · 2025-08-09
> > >
> > > We sincerely appreciate the reviewer's valuable comments and constructive suggestions, which have greatly improved the quality of our manuscript.

---

### Decision · Program_Chairs · 2025-09-17

**Decision:**

Accept (spotlight)

**Comment:**

The paper introduces Neighbor-aware Contrastive Disambiguation, a novel method for cross-modal hashing under redundant annotations. Key contributions: 1） It is a novel work to address redundant annotations in cross-modal hashing, a practical yet underexplored problem. 2）It combines Neighbor-aware Confidence Reconstruction for label disambiguation and Class-aware Robust Contrastive Hashing for robust retrieval. The experimental results show that the method outperforms baselines significantly, especially under high redundant supervision. Overall, the method is theoretically sound, empirically validated, and well-motivated. All reviewers recommend acceptance to the paper after the rebuttal. Therefore, the final decision is accepted.